# Morpho-Physiological Responses of Two Multipurpose Species from the Tropical Dry Forest to Contrasting Light Levels: Implications for Their Nursery and Field Management

**DOI:** 10.3390/plants11081042

**Published:** 2022-04-12

**Authors:** Erickson Basave-Villalobos, Víctor M. Cetina-Alcalá, Víctor Conde-Martínez, Miguel Á. López-López, Carlos Trejo, Carlos Ramírez-Herrera

**Affiliations:** 1Instituto Nacional de Investigaciones Forestales, Agrícolas y Pecuarias (INIFAP), Campo Experimental Valle del Guadiana, Carretera Durango-El Mezquital Km 4.5, Durango 34170, Mexico; 2Colegio de Postgraduados, Campus Montecilllo, Carretera México-Texcoco Km 36.5, Montecillo, Texcoco 56230, Mexico; vconde@colpos.mx (V.C.-M.); lopezma@colpos.mx (M.Á.L.-L.); catre@colpos.mx (C.T.); kmcramcolpos@gmail.com (C.R.-H.)

**Keywords:** agroforestry, ecological restoration, forest nurseries, indigenous tree species, reforestation

## Abstract

Understanding the responses that some plants exhibit to acclimatize and thrive in different light environments can serve as a guideline to optimize their production or establishment. Morpho-physiological changes in *Crescentia alata* and *Enterolobium cyclocarpum* were examined in response to varying light levels: 25%, 35%, 55% and 70% of photosynthetic photon flux density (PPFD) of total solar radiation. One-month-old seedlings were subjected to the light treatments; subsequently, the effects on morphology, photosynthetic capacity, nutrient status, non-structural carbohydrate reserves (NSC) and growth were evaluated in three-month-old seedlings. Light levels affected several morpho-physiological parameters. *C. alata* responded better to higher light levels and *E. cyclocarpum* to lower levels. Particularly, *C. alata* with 70% PPFD increased its size in height and diameter, and accumulated more biomass in leaves, stems, and roots; it also exhibited higher net assimilation rates, improved nitrogen and phosphorus status and growth. In contrast, *E. cyclocarpum* with 25% PPFD increased aboveground biomass, nitrogen levels and NSC in leaves. Both species show morpho-physiological changes that determine their ability to acclimatize to different light conditions. This serves as a basis for designing better management strategies in the nursery or field by defining the light environments conducive to a proper functioning.

## 1. Introduction

In Latin America, ecological restoration projects have been promoted in dry tropical forests due to the impacts of deforestation on these ecosystems [1]. To meet the objectives of these projects, it is essential to accelerate recovery processes in damaged areas according to their disturbance history [2]. In this regard, the active restoration approach, through forest plantations with native species involving reforestation and agroforestry activities using nursery plants is one of the most widely used strategies to accelerate the recovery of affected sites through the process of secondary succession [2,3].

However, the drawbacks that plantations frequently show are low survival and slow growth rates; hence, improving their performance continues to be a priority [4]. Inadequate species and site selection and failures in technologies for nursery production are two of the main technical problems that have received attention [5]. In the first case, species are often not ecologically feasible because they are not chosen according to the environmental conditions prevailing in each planting site [6,7], and, in the second case, appropriate nursery techniques are not employed to promote in the plants the morpho-functional characteristics that they require to thrive in the site conditions where they will be established [8].

In dry tropical forests, the seasonal dynamics of canopy openness and closure affect the light environment in the understory or forest floor [9]. This variability in light availability represents an environmental constraint for plant survival and growth, since this impacts plantations, whose performance depends on their ability to acclimatize to the prevailing environmental conditions and resources [7,8,10]. Moreover, the effect of such variability is intensified by the recurrent varying-scale clearings in turn due to various disturbance factors, including forest fires, changes in land use for agricultural activities and frequent exploitation practices for products such as firewood that constantly alter vegetation structure [11].

Given these environmental variations, based on the context of improving the performance of plantations, it is important to know the degree of acclimatization of the species used to select the most competent ones [12] and to define nursery techniques that promote appropriate morpho-physiological characteristics that are better matched to the irradiance levels prevailing in the planting sites [8]. 

In heterogeneous light environments, many plants are morphologically and physiologically modified at both leaf and whole plant levels to acclimate and optimize their photosynthesis and growth [12]. For example, if high irradiance prevails, leaf thickness or density is increased, and plant growth is adjusted to modify biomass allocation patterns in the photosynthetic apparatus and thus maintain carbon assimilation rates at adequate levels by minimizing photoinhibition [13,14,15]. Conversely, if the amount of available light is limited, then its interception, absorption and processing are optimized by increasing leaf area and reducing the costs of maintaining the photosynthetic apparatus [13,15]. Acclimation responses are species-dependent, and they vary as a function of the degree of phenotypic plasticity [16,17] or whether they are light-demanding or shade-tolerant species [18].

A documented experimental approach to determine the degree of acclimatization of forest species consists of analyzing the morphological and physiological changes of seedlings by exposing them to varying levels of light or shade [16]. In several species, this approach has led to important practical implications for improving their management in the nursery or in the field for production or establishment [17,18,19,20,21,22,23]. For example, a study with seedlings of *Tabebuia chrysotricha* (Mart. Ex DC.) Standl. defined that the species requires at least 50% of the total photosynthetically active radiation to increase its growth, morphological quality, and photosynthetic activity [21]. Conversely, for seedlings of *Carpotroche brasiliensis* (Raddi) A. Gray, it was determined that a shade level higher than 60% of full sunlight is optimal for the species [22], and unlike both species, in *Cedrela fissilis* Vell., it was found that seedlings are able to establish and thrive in environments with both high and low irradiance [23].

Based on this background, examining quantitively the acclimatation responses of a larger number of forest species to different levels of light deserves attention, because this approach gives useful insights that could optimize their management in the nursery or field. Particularly, many species of the dry tropics require this kind of study because much information on their propagation and management is lacking [24], which limits the available stock of these types of species in nurseries and plantation projects in the dry tropics where there is growing demand for this plant material. 

In the dry tropical regions of Latin America, *Crescentia alata* Kunth (Bignoniaceae) and *Enterolobium cyclocarpum* (Jacq.) Griseb. (Fabaceae) are two representative tree elements that have multiple uses [11,25]. To mention a few examples, these species are a source of timber products, have fruits that serve as food for humans, forage, and medicine, and provide ecosystem services such as shade, as well as the capacity—in the case of *E. cyclocarpum*—to fix atmospheric nitrogen [11,26]. Because they are multipurpose species and have shown adequate performance in growth and establishment, it has been recognized that both species are suitable for reforestation, restoration, and agroforestry activities in both open-field and enrichment plantings [11,26]. These two species serve as a study model to examine the experimental approach proposed above, since it is essential for them to define better management practices in nursery and field settings due to the wide variety of planting conditions in which they are being used [11,26]. This aspect is important to support experimentally since there are no previous records about the morphological and physiological responses that seedlings may have to variations in light. *C. alata* and *E. cyclocarpum* are heliophilous and dominate open fields in intermediate and adult stages, but in early stages, they can be found in different light environments due to the successional dynamics of dry tropical forests and their deciduous character, coupled with the alterations caused by human activities to the structure of this type of vegetation [9,11,26]. Therefore, the objective of this study was to analyze the changes in morphology, net assimilation rates, nutrient status, carbohydrate reserves and plant growth of both species growing in different light environments.

## 2. Results

Light levels (25% PPFD; 9.6 mol m^−2^ day^−1^, 35% PPFD; 13.5 mol m^−2^ day^−1^, 55% PPFD; 21.2 mol m^−2^ day^−1^, and 70% PPFD; 27 mol m^−2^ day^−1^), affected most of the variables measured in *C. alata* and *E. cyclocarpum* plants to different degrees. First, in *C. alata*, differences among light treatments were found in the following morphological variables: shoot height (SH), root-collar diameter (RCD), leaf biomass (LB), total biomass (TB), leaf biomass ratio (LBR), stem biomass ratio (SBR), root biomass ratio (RBR), leaf area (LA), specific leaf area (SLA), and leaf area ratio (LAR); and in the following physiological variables: net assimilation rate (NAR), nitrogen (N) and phosphorus (P) content, N and P nutrient uptake efficiency (NUE), and relative growth rate (RGR) (*p* < 0.05; Table 1 and Table 2). Most of these variables tended to increase their values as the amount of light increases; thus, 70% PPFD resulted in the maximum figures, while the other light treatments were associated with lower or intermediate values (Table 1 and Table 2). 

The trend of increasing values in the higher light treatment was not consistent in the biomass proportions (LBR, SBR and RBR) because of the similarity in the amounts of biomass allocated between the 70% and 25% PPFD plants. In both light levels, more than half of the plant biomass was allocated to the shoot (leaves and stem) and the rest to the root (Figure 1A). Similarly, SLA and LAR diverged from the noted trend because the highest values were obtained in the former with the 35% PPFD treatment, and in the latter, with 25% PPDF (Table 1). When comparing these values with the lower values registered by the plants under the 70% PPFD treatment, there was a difference of 23% and 16% for each variable, respectively (Table 1). 

Regarding N content, there was a 74% difference between the 70% and 35% PPFD treatments, which registered the highest and lowest values, respectively (Table 2). Similarly, P content in the 70% PPFD treatment was 69% higher than that in 25% PPFD treatment plants (Table 2). N uptake efficiency of the higher PPFD treatment (7%) was 77% higher than the lowest value registered for the 35% PPFD treatment (Table 2). Regarding P uptake efficiency, a difference of 69% was found between the 70% PPFD and 25% PPFD treatments, which reported the highest and lowest values, respectively (Table 2). Additionally, although non-structural carbohydrate (NSC) concentration showed a tendency to increase in both the lower and higher light treatments (with values ranging from 39 to 55 mg g^−1^ in leaves and 65 to 85 mg g^−1^ in roots), the variable did not show significant differences. Finally, RGR differed by 11% between the highest figure recorded with 70% PPFD and the lowest with 35% PPFD (Table 1).

For *E. cyclocarpum*, light levels significantly affected morphological variables: SH, root biomass (RB), LBR, SBR, RBR and LAR; as for physiological variables, there were differences in N concentration, N content and N uptake efficiency (*p* > 0.05; Table 3 and Table 4). Additionally, NSC concentration differed among treatments only in leaves (Figure 2).

The lower light treatment (25% PPFD) outperformed the highest one (70% PPFD) in terms of SH (Table 4). RB was 35% higher in the 55% PPFD treatment as compared with the 25% PPFD treatment (Table 3). As a result, plants in the lower light treatment (25% PPFD) allocated 73% of their biomass to aboveground organs; in contrast, in the higher light treatment (70% PPFD), the amount of biomass allocated to the aboveground was reduced by 11% and, hence, favored the biomass allocated to the root, up to almost 40% of the plant total biomass (Figure 1B). LAR increased 16% in plants with 25% PPFD compared to those with 70% PPFD, showing the highest and lowest values, respectively (Table 3). 

Values for N concentration, content, and uptake efficiency were higher in plants at 25% PPFD and decreased as the amount of light increased, with the lowest value being recorded for plants at 70% PPFD (Table 4). Differences between the 25% and 70% PPFD treatments ranged from 57% for concentration, 68% for content and 96% for nutrient uptake efficiency. 

Finally, NSC concentration was 27% higher in leaves of plants receiving 25% PPFD than in plants exposed to 70% PPFD; thus, carbohydrate concentration increased with light restriction, although the intermediate light treatments (35% and 55% PPFD) exhibited the lowest values (Figure 2). 

## 3. Discussion

The different light levels analyzed in this research affected the morphological characteristics, photosynthetic capacity, nutrient status, carbohydrate concentration and growth of *C. alata* and *E. cyclocarpum* plants. In heterogeneous light environments, plants of diverse species usually exhibit morphological and physiological modifications to efficiently use this resource according to their degree of phenotypic plasticity and acclimation capacity [14,15,27] thus, the changes that *C. alata* and *E. cyclocarpum* plants showed, mainly in the morphological variables measured, could be related to an acclimation capacity of these species to different light environments, used as a possible strategy to maintain adequate functioning [15]. In agreement with what was recorded in two *Dalbergia* species [28], *C. alata* and *E. cyclocarpum* plants at low light levels showed a plastic adjustment in the leaves to intercept more light by increasing their area, thereby affecting the values of specific leaf area and leaf area ratio [28], while at high light levels, leaves apparently developed a photo-protection strategy by increasing their density or thickness (lower specific leaf area) [29] and reducing the leaf area exposed to light [30]. According to dosAnjos et al. [31], morphological adjustments in leaves are one of the key functional responses that determine the photosynthetic plasticity of plants to light. Species with high photosynthetic plasticity have competitive advantages for light and other resources needed for photosynthesis against those with lower plasticity [29]. Likewise, the other morphological variables related to plant size (height and diameter) as well as biomass investment and allocation (LBR, SBR and RBR) expressed an adjustment capacity of plants to use light more efficiently when its availability was low. For example, *E. cyclocarpum* increased the height and promoted a greater biomass investment to the aboveground at the expense of that allocated to the root. In the face of light restriction, this response suggests a mechanism to increase the size of the photosynthetic apparatus, thereby improving light acquisition [32]. This behavior coincides with that reported in *Tabebuia chrysotricha* (Mart. Ex DC.) Standl. [21] and in *Prosopis laevigata* (Humb. & Bonpl. ex Will.) M. C. Johnst. [33].

In *C. alata*, high light levels favored its growth; hence, with 70% PPFD, the plants increased their values in most morphology variables at leaf, stem, and root level; in contrast, low light significantly reduced its growth. Another *Crescentia* species (*Crescentia cujete* L.) also displayed a behavior similar to that of *C. alata* in a study comparing the performance of the species in a high- and low-light environment [34]. Shade is considered a stress that limits photosynthesis and plant growth [14], to the extent that plants only produce a fraction of total biomass compared to those growing in higher light environments [15].

For *C. alata*, the 70% PPFD treatment promoted higher net assimilation rates that explain, in part, the higher growth rates in this light condition. Khurana and Singh [35] report that forest species in the dry tropics achieve high growth and net assimilation rates at high light levels. This idea is supported by the results of the present study and by findings presented for *Enterolobium contortisiliquum* (Vell.) Morong [36]. The ability of light-demanding species to maintain high rates of growth and photosynthesis is often attributed to their ability to maintain a positive carbon balance under conditions of high solar radiation, for which they execute mechanisms that avoid photoinhibition and oxidative stress [15,37]. It would be interesting for further studies to investigate this issue, based on the approach proposed by Calzavara et al. [12] and Cerqueira et al. [22], in order to elucidate in greater detail the photosynthetic plasticity and photoinhibition vulnerability of *C. alata* and *E. cyclocarpum*, involving plants growing under full sunlight treatment, a condition that was not possible to explore in this study, because the experiment was conducted in a greenhouse whose film restricted full sun exposure. Additionally, given the fact that this study was carried out under artificial conditions that were not capable of simulating fluctuations in irradiance as naturally occurs in the understory, it is recommended that further studies on this topic consider the influence of light fluctuations, such as those generated due to sunflecks, in order to better understand all the possible responses of acclimation and their interactions or trade-offs that plants undergo in a fluctuating irradiance environment, because this represents a constraint to which plants should modulate themselves with a set of morphological and physiological features, depending on the prevailing environment, so that they will be able to attain maximum photosynthetic rates and appropriately adjust other plant functions [13,15,38,39]. Possibly, in face of the methodological limitation that arguably prevented evaluation under conditions similar to those occurring in nature, the studied plants did not exhibit all of their acclimation potential under the variety of light conditions tested.

On the other hand, responses in nutrient status because of light levels can generally be explained by the influence that light has on the availability of other resources—in this case, nutrients [40]. In *C. alata* plants, as the amount of light increased, the content and uptake efficiency of nitrogen and phosphorus increased, to the point that at 70% PPFD, the plants had the best nutrient status for both elements. This response can be explained by the relationship that exists between nutrient and carbon assimilation; specifically, because light was not a limiting factor in the 70% PPFD treatment, there was sufficient energy to increase carbon gains and thus acquire the nutrients that usually represent a high energy cost for plants when they experience resource limitations such as light [41].

Hiremath [42] and Berendse et al. [43] argue that there is a close relationship between the assimilation of nutrients and carbon through photosynthesis, such that the acquisition of nutrients, their transport within plants and their assimilation into organic compounds are determined by the efficiency with which plants use nutrients for photosynthesis. For *E. cyclocarpum*, nutrient status responses were different from those of *C. alata*. Nitrogen status improved not at higher light levels but at lower levels. The higher values in concentration, content and uptake efficiency found in plants growing only with 25% PPFD suggest that in high light, the photosynthetic capacity of plants was probably limited by possible photoinhibition effects, and, therefore, the ability to assimilate nutrients was affected, based on the evidence provided by Santos et al. [8] with plants of *Chrysophyllum sanguinolentum* (Pierre) Baehni.

Another possible explanation is that the amount of light probably affected plant water status. Water losses by transpiration could have been higher in those plants subjected to higher levels of radiation. Given that nutrient assimilation in plants also depends on water status [44], it is likely that plants exposed to a higher amount of light suffered episodes of water stress, which limited their ability to efficiently use the nutrients supplied. Although efforts were made to maintain the same moisture content in the growing media through frequent and uniform irrigations, perhaps the plants with greater shading did not experience water deficit, and this accordingly allowed them to achieve better efficiency in their use of the fertilizer. Finally, a third hypothesis is that under shaded conditions, the plants increased their chlorophyll content because that increases their efficiency in capturing light when its availability is low [45], and, because of this, it was a priority for investments of higher concentrations of nitrogen as the amount of light received by the plants was lower. This explanation is corroborated by higher values in the specific leaf area and leaf area ratio of plants exposed to greater shading, as they are associated with an increase in chlorophyll content [46].

In relation to carbohydrates, the higher concentration in leaves of *E. cyclocarpum* plants that grew under the lower light levels indicates that this condition favored the accumulation and storage of these assimilates. Shade stimulates the accumulation of carbohydrates in the reserve organs because this condition can be a stress factor that plants must overcome by maintaining a positive carbon balance. This condition is achieved by increasing the amount of reserves. More reserves can be generated when carbon gains exceed the demands represented by other plant functions that compete for resources with storage, such as growth [47]. Storing carbohydrate reserves increases the probability of plant survival during periods of stress. In plantations, this occurs mostly at the time of transplanting. The response in carbohydrate increase has important implications for nursery plant production. Light management can be implemented as a cultural practice to manipulate the physiological quality of plants, since the amount of carbohydrates is a decisive attribute in the performance of plantations [48]. 

Finally, when comparing the responses of *C. alata* versus *E. cyclocarpum*, it was noticeable that these species differ in their light requirement. *E. cyclocarpum* grew better in the lower light levels as opposed to *C. alata*, which preferred the higher level (70% PPFD). This divergence coincides with that reported for *Dalbergia nigra* and *Dalbergia miscolobium*, two species that, despite belonging to the same genus, differ in their light requirements [28]. These differences are also influenced by other environmental factors that affect growth but are ultimately regulated by the light environment in which the plants grow [14]. Accordingly, it is crucial to provide each species with the light environment in which it performs best. The results obtained have important practical implications. Nursery or field management of *E. cyclocarpum* can be optimized in environments with greater shade than those required for *C. alata*, so nursery production techniques and planting schemes for each species should be adjusted to these light preferences.

## 4. Materials and Methods

### 4.1. Study Area

The experiment was carried out under greenhouse conditions in the facilities of the forest nursery of the Forest Science Graduates Program of Colegio de Postgraduados Campus Montecillo. The greenhouse was covered with a transparent 800-gauge plastic film with a transmissivity coefficient of 70%.

### 4.2. Plant Material

*C. alata* and *E. cyclocarpum* plants were produced from seeds in a container system. Seeds were collected from scattered trees in the Tierra Caliente region of the state of Guerrero, Mexico. Seeds were sown in 25-cavity rigid polypropylene trays set on metallic tables. Pots 310 and 380 mL in volume were used for *C. alata* and *E. cyclocarpum*, respectively. The pots were filled with a potting mix consisting of peat moss, perlite, and vermiculite (2:1:1 *v*/*v*) amended with a controlled-release fertilizer (Multicote 8^®^ 18 N:6 P_2_O_5_:12 K_2_O + 2 MgO + ME; Haifa Chemicals Ltd, Haifa, Israel) at a dosage of 6 g L^−1^. Direct seeding was performed in May 2019 with homogeneously sized seeds and no apparent insect or fungal damage, following the recommendations of Valverde-Rodríguez et al. [49] for *C. alata* and Viveros-Viveros et al. [50] for *E. cyclocarpum*. *E. cyclocarpum* seedlings emerged in the first week after sowing, and *C. alata* seedlings in the second week. 

### 4.3. Light Levels and Experimental Design

One-month-old seedlings were subjected to four sunlight treatments in the greenhouse: 25%, 35%, 55% and 70% photosynthetic photon flux density (PPFD) with respect to total photosynthetically active radiation, in a completely randomized block experimental design with four replicates. Each experimental unit consisted of 16 plants. The blocking criterion was the temperature gradient inside the greenhouse.

Seventy percent PPFD corresponded to the high light treatment. This light level corresponded to the transmissivity of the plastic film. The remaining PPFD treatments of light attenuation (25%, 35% and 55%) were manipulated with black monofilament shade nets commonly used in horticulture. These light treatments were chosen because they cover the range of understory irradiance that usually prevails in the dry forest during the wet season according to data of Lebrija-Trejos et al. [9]. Prior to establishment of the trial, the percentage of shade was verified by simultaneous midday measurements of PPFD (µmol m^−2^ s^−1^) above and below the shade nets, carried out with an AccuPAR Ceptometer model LP-80 (Decagon Devices, Inc., Washington, DC, USA). Dome-like canopies were constructed with the shade nets for each experimental unit according to the corresponding PPFD treatment. Maximum and minimum temperature (°C) and relative humidity (%) were monitored daily at midday in each condition with a TER-150 Steren^®^ digital thermohygrometer (Electrónica Steren S. A. de C. V., Mexico). Additionally, as recommended by Poorter et al. [16], the daily light integral (mol m^−2^ day^−1^) was calculated according to the procedure described by Torres and Lopez [51], for which PPFD was measured daily every hour during the entire study period with a LightScout^®^ quantum PAR sensor connected to a WatchDog^®^ micro station (Spectrum Technologies, Inc., Aurora, IL, USA). Table 5 lists the average values obtained for each light condition. 

### 4.4. Plant Sampling, Morphological and Physiological Measurements

Sampling was carried out on three-month-old plants in a set of 12 individuals randomly selected from each block and light treatment. The plants were extracted from the pots, and the growing media were carefully removed from the root plugs with tap water to be destructively sampled. They were separated into leaves, stems, and roots; then, morphological variables such as shoot height (SH; cm), root collar diameter (RCD; mm), and leaf area (LA; cm^2^) were measured, the latter with a LI-3100C leaf area meter (LI-COR Inc., Lincoln, NE, USA). Subsequently, each sample was rinsed in deionized water, oven-dried for 72 h at 70 °C (until constant weight) and weighed for biomass measurements. The biomass (g) of leaves (LB), stem (SB), root (RB) and total (TB) was quantified with an analytical balance (AND^®^ GR-120, A&D Company, Ltd. Tokyo, Japan). The biomass values were used to determine the ratios of leaf biomass (LBR; leaf biomass/total biomass), stem (SBR; stem biomass/total biomass) and root (RBR; root biomass/total biomass). Likewise, specific leaf area (SLA; cm^2^ g^−1^) and leaf area ratio (LAR; cm^2^ g^−1^) were calculated using the values of leaf area and leaf biomass for the first variable, and considering the leaf area and total biomass for the second. 

Regarding physiological variables, net assimilation rates (NAR; mg cm^−2^ day^−1^) were determined as a measure of the photosynthetic capacity of the plants because this is a complex physiological parameter associated with photosynthetic and respiration rates [52]. Additionally, the status of nitrogen (N), phosphorus (P) and potassium (K), as well as the non-structural carbohydrate concentration (NSC), sugars and starch, were also quantified. 

NAR was calculated according to the following formula:NAR = [(TB_2_ − TB_1_)∗(lnLA_2_ − lnLA_1_)]/[(LA_2_ − LA_1_)∗(T_2_ − T_1_)](1)
where TB refers to total plant biomass, LA to leaf area of plants in absolute and logarithmic values (ln), and T to time. In all cases, 1 and 2 refer to an initial and final harvest, respectively, 60 days apart. 

Chemical analyses of whole plants were conducted to evaluate the status of N, P and K expressed as concentration (%) and nutrient content (mg plant^−1^). N concentration was determined by the micro-Kjeldahl method; P concentration by colorimetric determination with yellow vanadate-molybdate complex, and K concentration by flame emission spectrophotometry. The nutrient content (N, P, K) was calculated by multiplying the nutrient concentration by biomass values of the plants. Likewise, nutrient uptake efficiency (NUE) was determined from the following equation [53]:NUE = [Plant nutrient content (PNC) − (PNC of unfertilized seedlings)]/Nutrient fertilization rate∗[100](2)

The nutrient content of unfertilized seedlings was quantified for newly emerged seedlings. The nutrient fertilization rate (N, P, K) was computed per plant based on the amount of the fertilizer applied and the nutrient concentration labeled in the fertilizer Multicote 8^®^ 18-6-12. 

The NSC concentration was estimated by the anthrone method on leaves and roots per quadruplicate. Sugars were extracted with 80% ethanol (EtOH) using Soxhlet equipment. Quantification of total sugars was performed by the anthrone–sulfuric acid method with a colorimetric determination at 625 nm by using a UV/visible spectrophotometer (Jenway™ 6305, Cole-Parmer Ltd., Stone, Staffordshire, UK). Calculations of total sugars (mg g^−1^ fresh weight) were performed based on a glucose calibration curve (2.5 mg mL^−1^) over a concentration range of 10 to 150 μg, as described by Hansen and Møller [54]. 

Finally, growth (biomass increase at whole plant level) was measured in terms of relative growth rates (RGR; mg g^−1^ day^−1^). RGRs were calculated as follows [55]:RGR = (lnTB_2_ − lnTB_1_)/Δt (days)(3)
where TB 1 and 2 refers to the plant total biomass in logarithmic values (ln) in the initial and final harvest, respectively. Δt is the interval of time between harvests (60 days).

The initial harvest required to estimate the NAR and RGR variables was conducted at the beginning of the light treatments (June 2019).

### 4.5. Data Analyses

The effect of the light level on the variables evaluated was assessed by a one-way ANOVA at a significance level of α = 0.05, after validation of the assumptions of homoscedasticity and normality. When ANOVA assumptions were not met, data were transformed using logarithm, inverse, or square root functions. When significant effects were detected (*p* < 0.05), means were compared by using Fisher’s least significant difference (LSD) test at α = 0.05. Statistical analyses were conducted with InfoStat version 2008 [56].

## 5. Conclusions

The morphological and physiological changes in the seedlings of each species determine their ability to acclimatize to different light conditions. However, these changes are favored in the case of *C. alata* with higher light levels than those required by *E. cyclocarpum*. This highlights the importance of promoting in the nursery or field the light environment in which each species performs satisfactorily. For *E. cyclocarpum* and *C. alata*, that corresponded to 25% and 70% of the photosynthetic photon flux density, respectively. The implications from these results will contribute to the formulation of better seedling production and management strategies, both in the nursery and in the field, for the different planting schemes in which these species are used in the dry tropics.

## Figures and Tables

**Figure 1 plants-11-01042-f001:**
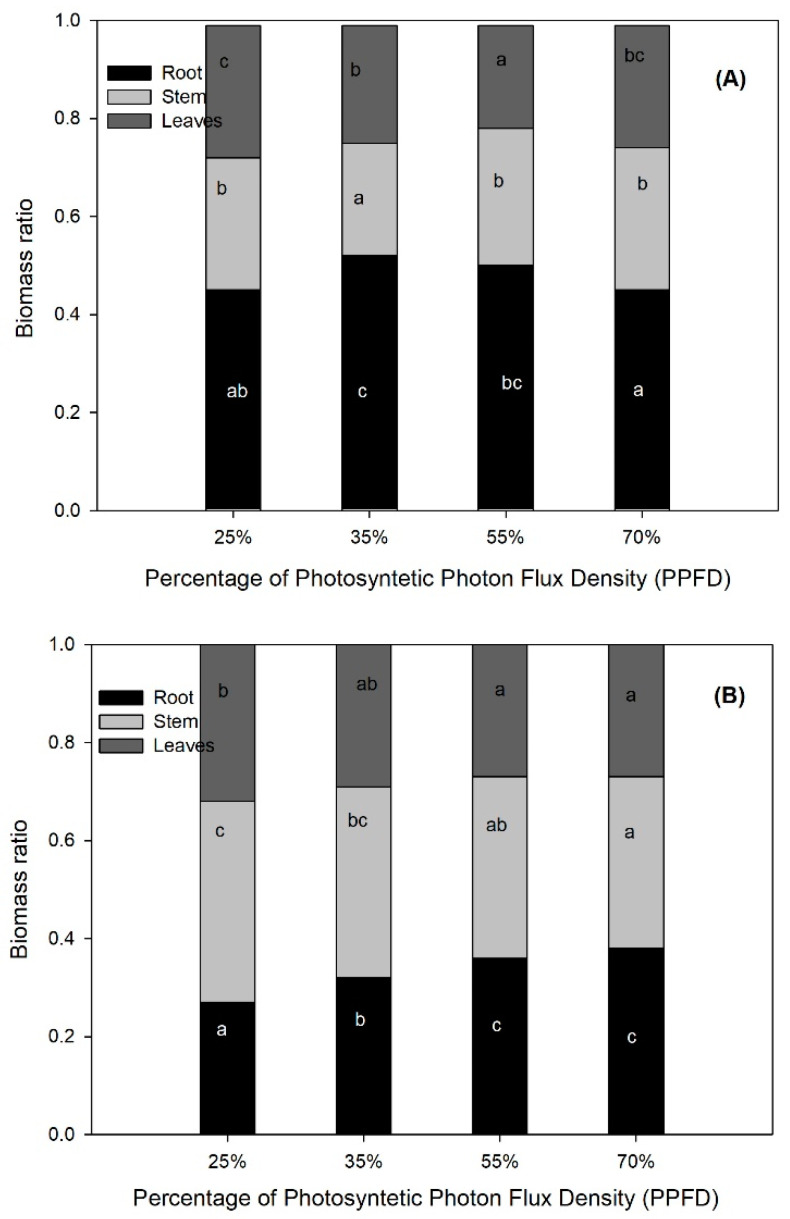
Biomass allocation patterns of (**A**) *Crescentia alata* and (**B**) *Enterolobium cyclocarpum* plants exposed to varying photosynthetic photon flux density (PPFD) levels (*n* = 12). Segments in the bars with a letter in common among bars are not significantly different (*p* > 0.05) according to Fisher’s LSD test (α = 0.05).

**Figure 2 plants-11-01042-f002:**
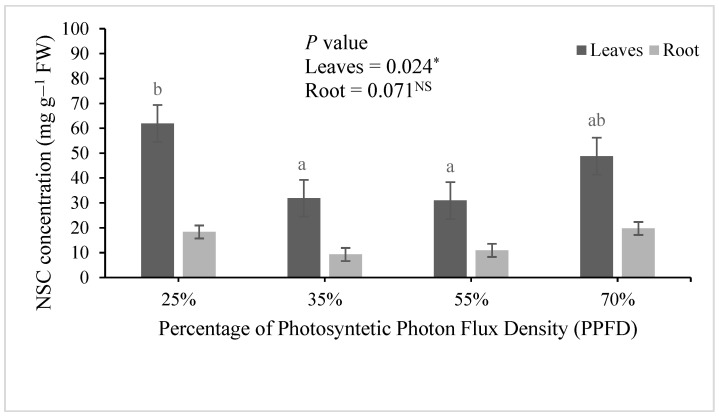
Mean values (±standard deviation) of non-structural carbohydrate concentration (NSC) determined in plants of *Enterolobium cyclocarpum* exposed to varying photosynthetic photon flux density (PPFD) levels (*n* = 4). FW, fresh weight. Bars with a letter in common are not significantly (NS) different (*p* > 0.05) according to Fisher’s LSD test. (α = 0.05). *, significant; ^NS^, non-significant.

**Table 1 plants-11-01042-t001:** Statistical significance and averages (*n* = 12) in morphology variables, net assimilation rate and growth of *Crescentia alata* plants exposed to varying light levels in relation to photosynthetic photon flux density (PPFD). For the same variable, means with letters in common are not significantly different (*p* > 0.05) according to Fisher’s LSD test (α = 0.05). SH, shoot height; RCD, root-collar diameter; LB, leaf biomass; SB, stem biomass; RB, root biomass; TB, total biomass; LA, leaf area; SLA, specific leaf area; LAR, leaf area ratio; NAR, net assimilation rate; RGR, relative growth rate.

Variable	*p* Value	Light Level (PPFD; %)			
		25	35	55	70
SH (cm)	0.0001	10.88 b	8.71 a	10.75 b	12.29 c
RCD (mm)	0.0389	4.85 a	5.19 ab	5.15 ab	5.75 b
LB (g)	0.0015	0.52 a	0.43 a	0.47 a	0.63 b
SB (g)	0.0001	0.52 ab	0.42 a	0.59 b	0.71 c
RB (g)	0.0918	0.89	0.93	1.06	1.12
TB (g)	0.0030	1.93 a	1.78 a	2.12 ab	2.45 b
LA (cm^2^)	0.0015	146.72 a	134.01 a	139.09 a	163.23 b
SLA (cm^2^ g^−1^)	0.0203	287.41 ab	325.16 b	313.84 b	264.48 a
LAR (cm^2^ g^−1^)	0.0038	77.71 b	77.28 b	67.27 a	67.21 a
NAR (mg cm^−2^ day^−1^)	0.0044	0.14 ab	0.14 a	0.16 bc	0.17 c
RGR (mg g^−1^ day^−1^)	0.0055	26.88 a	26.14 a	27.59 ab	28.94 b

**Table 2 plants-11-01042-t002:** Statistical significance and mean values (*n* = 4) of variables related to nutrient status of *Crescentia alata* plants exposed to varying photosynthetic photon flux density (PPFD) levels. For a particular nutrient in the same variable, means with a letter in common are not significantly different (*p* > 0.05) according to Fisher’s LSD test (α = 0.05).

Light Level (PPFD; %)	Nutrient Concentration (%)			Nutrient Content (mg plant^−1^)			Nutrient Uptake Efficiency (%)		
	N	P	K	N	P	K	N	P	K
25	2.08	0.26	0.90	40.08 ab	4.99 a	13.63	11.04 ab	4.45 a	6.08
35	1.88	0.29	0.71	33.43a	5.08 a	14.69	9.05 a	4.53 a	6.55
55	1.88	0.28	0.83	39.82 ab	5.90 ab	15.00	10.96 ab	5.26 ab	6.69
70	2.38	0.34	0.71	58.27 b	8.45 b	22.08	16.47 b	7.54 b	9.86
*p* value	0.05	0.103	0.306	0.032	0.043	0.086	0.032	0.043	0.086

**Table 3 plants-11-01042-t003:** Statistical significance and averages (*n* = 12) in morphology variables, net assimilation rate and growth of *Enterolobium cyclocarpum* plants exposed to varying photosynthetic photon flux density (PPFD) levels. For the same variable, means with letters in common are not significantly different (*p* > 0.05) according to Fisher’s LSD test (α = 0.05). SH, shoot height; RCD, root-collar diameter; LB, leaf biomass; SB, stem biomass; RB, root biomass; TB, total biomass; LA, leaf area; SLA, specific leaf area; LAR, leaf area ratio; NAR, net assimilation rate; RGR, relative growth rate.

Variable	*p* Value	Light Level (PPFD; %)			
		25	35	55	70
SH (cm)	<0.0001	50.75 c	45.00 b	40.08 ab	36.25 a
RCD (mm)	0.1977	5.17	5.65	5.40	5.35
LB (g)	0.3821	1.48	1.37	1.33	1.22
SB (g)	0.2028	1.89	1.86	1.78	1.52
RB (g)	0.0042	1.24 a	1.48 ab	1.68 b	1.60b
TB (g)	0.7046	4.62	4.70	4.79	4.33
LA (cm^2^)	0.3821	220.31	203.12	198.21	181.02
SLA (cm^2^ g^−1^)	0.3220	148.58	148.71	148.76	149.36
LAR (cm^2^ g^−1^)	0.0445	47.37 b	43.07 ab	41.17 a	40.73 a
NAR (mg cm^−2^ day^−1^)	0.1278	0.39	0.42	0.43	0.40
RGR (mg g^−1^ day^−1^)	0.4892	19.84	19.92	20.03	18.88

**Table 4 plants-11-01042-t004:** Statistical significance and mean values (*n* = 4) in variables related to nutrient status of *Enterolobium cyclocarpum* plants exposed to varying photosynthetic photon flux density (PPFD) levels. For a particular nutrient in the same variable, means with a letter in common are not significantly different (*p* > 0.05) according to Fisher’s LSD test. (α = 0.05).

Light Level (PPFD; %)	Nutrient Concentration (%)			Nutrient Content (mg)			Nutrient Uptake Efficiency (%)		
	N	P	K	N	P	K	N	P	K
25	2.92 b	0.27	0.95	134.75 b	12.65	44.07	27.14 b	7.57	14.06
35	2.23 ab	0.27	0.82	104.73 ab	12.57	38.58	19.82 ab	7.51	12.05
55	1.93 a	0.27	0.80	92.21 a	12.98	38.41	16.77 a	7.81	11.99
70	1.86 a	0.30	0.94	80.32 a	13.04	40.59	13.88 a	7.86	12.79
*p* value	0.024	0.644	0.281	0.018	0.826	0.546	0.018	0.826	0.546

**Table 5 plants-11-01042-t005:** Environmental conditions recorded in each light treatment.

Environmental Variable	Light Level (PPFD; %)				
	25	35	55	70	Open Sky *
Maximum temperature (°C) ^¶^	36.0 ± 3.1	36.4 ± 3.0	36.6 ± 3.2	42.6 ± 4.5	-
Minimum temperature (°C) ^¶^	15.9 ± 1.6	14.1 ± 1.5	15.5 ± 1.7	14.8 ± 1.8	-
Relative humidity (%) ^¶^	27.4 ± 6.4	28.3 ± 8.5	28.5 ± 8.1	26.5 ± 6.2	-
Daily light integral (mol m^−2^ day^−1^)	9.6	13.5	21.2	27.0	38.6

^¶^ Average values of daily maximums and minimums over a 24 h period recorded at 12:00 p.m. during the entire study period. * Recorded for comparison.

## Data Availability

The data presented in this study are available on request from the corresponding author.

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
