# Peer review of "Morpho-Physiological Responses of Two Multipurpose Species from the Tropical Dry Forest to Contrasting Light Levels: Implications for Their Nursery and Field Management"

_plants, 2022, doi:10.3390/plants11081042_

Round 1

Reviewer 1 Report

Dear Authors,

Reviewer comments plants-1659429

The manuscript entitled „Morpho-physiological responses of two multipurpose speciesfrom the tropical dry forest to contrasting light levels: implications for their nursery and field management“ represents a useful study aimed at an investigation of the effects of reduced photosynthetic photon flux density (PPFD; 25, 35, 55 and 70% PPFD) of total solar radiation on morphophysiological characteristics of young plants Crescentia alata and Enterolobium cyclocarpum native to tropical dry forests.

I have some comments on the present manuscript which have to be addressed by the authors:

Terminology:

The abbreviations used for the morphophysiological characteristics have to be explained directly in the text when used for the first time or they have to be explained in a separate Abbreviations list.

The term „non-structural carbohydrates NSC“ has to be defined. Which carbohydrates belong to „non-structural carbohydrates“ mentioned in the text??

The difference between „N concentration“ and „N content“ has to be clearly defined in Results section when these terms are sued for the first time.

Materials and methods, Results:

In Table 5, the authors finally provide the data of absolute values for PPFD expressed as daily light integral (mol m-2day-1) (not mol m-2dia-1 = Spanish!!) for the individual light treatments of 25, 35, 55 and 70% PPFD reduction. I recommend the authors to add the PPFD value expressed as daily light integral for the full PPFD, i.e., 100% PPFD in Table 5. Moreover, although it is methodology, I recommend the authors to provide the data on absolute PPFD values expressed as daily light integral in (mol m-2day-1) at the beginning of Results section to provide these crucial data to the readers at the beginning of the data description.

Results, the data presented in Figures andTables: In the corresponding figure and table legends as well as in Materials and methods, section on Statistical analysis, the information on the number of biological replicates used for the determination of morphophysiological characteristics has to be added. Moreover, regarding the bars in the graphs, do they represent SD or SE values??

Formal comments:

Introduction, line 104: Add a comma following the words „…but in early stages,…“

Results, line 114: Add a comma following the words: „First, in C. alata,…“

Results, line 118: Replace the word „with“ with „in“ in the statement „…thus, 70% PPFD resulted in the maximum figures…“

Results, line 130: Add a comma following the words: „In both light levels,…“

Results, line 149: Add a comma following the words „Regarding P uptake efficiency,….“

Results, line 176: Add a comma following the words „…in contrast, in the higher light treatment (70% PPFD),….  

Results, line 179: Add „the“ preceding the words „lowest values“ in the statement „…which showed the highest and the lowest values, respecively…“

Results, Figure 2 legend, line 197: Remove an extra word „plants“ in the statement „Non-structural carbohydrates concentration (NSC) determined in plants of Enterolobium cyclocarpum  exposed to varying photosynthetic photon flux density…“

Discussion, line 207: Correct the term „acclimation“, not „acclimatation“.

Discussion, line 223: Divide the very long sentence into the two shorter ones: „For example, E. cyclocarpum increased the height and promoted a greater biomass investment to the aboveground at the expense of that allocated to the root.“ – the first sentence „In the face of light restriction, this is a mechanism to increase the size of the photosynthetic apparatus and thereby improving light acquisition…“

Discussion, line 227: Add a comma following the words „…hence with 70% PPFD,…“

Discussion, line 251: Add a comma following the words „…to the point that at 70% PPFD,..“

Discussion, line 264: Add a comma following the words „…suggest that in high light,…“

Materials and methods, line 366: Add the word „one“ following the words „the first“ in the statement „….the first one using the values of leaf area and leaf biomass,…“

Materials and methods, line 369: Correct the term „anthrone method“,not „Anthrona method.“

Final recommendation: Accept after a major revision.

Author Response

REVIEWER 1

Terminology:

Reviewer comment:The abbreviations used for the morphophysiological characteristics have to be explained directly in the text when used for the first time or they have to be explained in a separate Abbreviations list.

Response: We appreciate the suggestion. This information is provided as requested.

Reviewer comment:The term „non-structural carbohydrates NSC“ has to be defined. Which carbohydrates belong to „non-structural carbohydrates“ mentioned in the text?

Response: We define within the text that non-structural carbohydrates refer to sugars and starch.

Reviewer comment: The difference between „N concentration“ and „N content“ has to be clearly defined in Results section when these terms are sued for the first time.

Response: We appreciate the observation; however, we considered that no changes are needed because we differentiate nutrient concentration results from those of content.

Materials and methods, Results:

Reviewer comment: In Table 5, the authors finally provide the data of absolute values for PPFD expressed as daily light integral (mol m-2day-1) (not mol m-2dia-1 = Spanish!!) for the individual light treatments of 25, 35, 55 and 70% PPFD reduction. I recommend the authors to add the PPFD value expressed as daily light integral for the full PPFD, i.e., 100% PPFD in Table 5. Moreover, although it is methodology, I recommend the authors to provide the data on absolute PPFD values expressed as daily light integral in (mol m-2day-1) at the beginning of Results section to provide these crucial data to the readers at the beginning of the data description.

Response: Thank you again for this feedback. First, we have translated the Spanish word “Día”. Second, we include in the table the value of Daily light integral (DLI) registered for open-sky condition. Finally, we include at the beginning of result the values of DLI for each light treatment as requested by the reviewer.  

Reviewer comment: Results, the data presented in Figures and Tables: In the corresponding figure and table legends as well as in Materials and methods, section on Statistical analysis, the information on the number of biological replicates used for the determination of morphophysiological characteristics has to be added. Moreover, regarding the bars in the graphs, do they represent SD or SE values??

Response: In the corrected version of the manuscript, we show the information on the number of biological replicated or sample size (n), in addition to the information provided in the original manuscript.

Finally, some other corrections were suggested; mostly signs of punctuation, all were solved.

Reviewer 2 Report

Title: Morpho-physiological responses of two multipurpose species from the tropical dry forest to contrasting light levels: implications for their nursery and field management

Authors: Basave-Villalobos et al.

Comments to the author(s).

In this study, the authors have performed field experiments comparing physiological responses woody species to different light environments. Although I admit this topic is an important issue and that the dataset presented in the manuscript may be of potential values, I also found several major flaws in the presentation of their study. Unfortunately, therefore, in my opinion the manuscript in its current form is not suitable for publication in high-impact journals such as Plants (IF = 3.9).

General issues

(1) Novelty of the present study is not clarified. The data presented here is in fact comparison of sun- vs shade-grown plants, which has been extensively studied (see literatures listed below). I think that recent (> year 2015) were not well covered in this manuscript, especially in the Introduction and Discussion chapters.

(Leaf-level responses)

Poorter, H.; Niinemets, U.; Ntagkas, N.; Siebenkas, A.; Maenpaa, M.; Matsubara, S.; Pons, T. A meta-analysis of plant responses to light intensity for 70 traits ranging from molecules to whole plant performance. New Phytol. 2019, 223, 1073–1105

dos Santos, V.A.H.F.; Ferreira, M.J. Are photosynthetic leaf traits related to the first-year growth of tropical tree seedlings? A light-induced plasticity test in a secondary forest enrichment planting. For. Ecol. Manag. 2020, 460, 117900

Ntawuhiganayo, E.B.; Uwizeye, F.K.; Zibera, E.; Dusenge, M.E.; Ziegler, C.; Ntirugulirwa, B.; Nsabimana, D.; Wallin, G.; Uddling, J. Traits controlling shade tolerance in tropical montane trees. Tree Physiol. 2020, 40, 183–197.

Deguchi, R.; Koyama, K. Photosynthetic and Morphological Acclimation to High and Low Light Environments in Petasites japonicus subsp. giganteus. Forests 2020, 11, 1365. https://doi.org/10.3390/f11121365

Kitajima, K. Relative importance of photosynthetic traits and allocation patterns as correlates of seedling shade tolerance of 13 tropical trees. Oecologia 1994, 98, 419–428.

Kupers, S.J.; Wirth, C.; Engelbrecht, B.M.J.; Hernández, A.; Condit, R.; Wright, S.J.; Rüger, N. Performance of tropical forest seedlings under shade and drought: An interspecific trade-off in demographic responses. Sci. Rep. 2019, 9, 18784.

(Whole-plant level morphological responses, such as LAR)

Ntawuhiganayo et al. (listed above)

Poorter et al. (2019) listed above.

Gruntman, M.; Segev, U.; Tielbörger, K.; Gange, A. Shade-induced plasticity in invasive Impatiens glandulifera populations. Weed Res. 2019, 60, 16–25.

They mentioned that it is interesting to study additional species:

(L85-) Based on this background, it is interesting to examine the responses of a larger number of forest species to different levels of light in order to optimize their management in the nursery or field.

That is merely their opinion, and the reader (including me) do not think the present manuscript is interesting. (and I rarely found an expression like “it is interesting” in high-impact journals). The reason of this is that they do not explain what is the novel and important points of their study. Every dataset is novel, but they should explain why it is important and novel. This point is particularly important to be published in high-impact journals.

Additionally, I wonder whether there are really no study on the sun vs shade acclimation of the two species (C. alata and E. cyclocarpum).

(2) Methodology is poorly described.

They set PPFD levels (25, 35, 55, 70) using films. However, if my understandings is correct, these values are the transmissivity of the plastic film, which may not necessary matches the actual PPFD experienced by the study plants. They mentioned that they have actually measured PPFD. If so, they should show the actual rPPFD values experienced by the plants.

(Table 5) Information on daily integral of outside (open-sky condition) should also be provided.

(3) In several recent studies it has been recognized that using shade cloth or film may not always mimic the natural light environment in shaded understory, because shade film or cloth cannot produce sunflecks (see literatures listed below). This fact has long been recognized (e.g., Valladares et al. 2008; Poorter et al. 2016; Morales & Kaiser 2020), and recent studies show quantitative evidence (Deguchi & Koyama 2020, listed above). I think the data presented here is potentially valuable, but I recommend the authors to clarify the limitation of the present results obtained from artificial shading experiment, with an argument on this point.

Valladares, F.; Zaragoza-Castells, J.; Sanchez-Gomez, D.; Matesanz, S.; Alonso, B.; Portsmuth, A.; Delgado, A.; Atkin, O.K. Is shade beneficial for Mediterranean shrubs experiencing periods of extreme drought and late-winter frosts? Ann. Bot. 2008, 102, 923–933.

Poorter, H.; Fiorani, F.; Pieruschka, R.; Wojciechowski, T.; van der Putten, W.H.; Kleyer, M.; Schurr, U.; Postma, J. Pampered inside, pestered outside? Differences and similarities between plants growing in controlled conditions and in the field. New Phytol. 2016, 212, 838–85

Morales, A.; Kaiser, E. Photosynthetic acclimation to fluctuating irradiance in plants. Front. Plant Sci. 2020, 11, 268, https://doi.org/10.3389/fpls.2020.00268

Specific comments

(In Tables) These units should read “day-1” (not “dia”)

It seems that there are several studies on the same two species (example shown below); they should be checked:

Foroughbakhch et al. Forest Ecology and Management Volume 235, Issues 1–3, 1 November 2006, Pages 194-201, https://doi.org/10.1016/j.foreco.2006.08.012

Author Response

REVIEWER 2

Reviewer comment: Novelty of the present study is not clarified. The data presented here is in fact comparison of sun- vs shade-grown plants, which has been extensively studied (see literatures listed below). I think that recent (> year 2015) were not well covered in this manuscript, especially in the Introduction and Discussion chapters.

Response: We appreciate the feedback. Overall, this study proposes as novelty, an experimental approach focused on understanding the morpho-physiological responses of two native tree species (Crescentia alata and Enterolobium cyclocarpum) to heterogenous light environments, that gives useful insights needed for optimizing their nursery production or field establishment. Certainly, although the acclimation responses of plants to different light environments have been examined in several species, we, thanks to those backgrounds, identified a gap of knowledge, on the one hand, in relation to examining the particular responses of the species which are socioeconomic and ecologically important in the dry tropics; on the other hand, in generating, through a quantitative evaluation of the changes in several morphological and physiological parameters, applied knowledge whose implications can be  relevant either to improve nursery production systems of this kind of species or to increase plantation survival of reforestations in the dry tropics. We consider that the aim and scope is clarified, and we provide some other current references to support our research approach.

  • Moretti, A.P.; Olguin, F. Y.; Pinazo, M.A.; Graciano, C. Water and light stresses drive acclimation during the establishment of a timber tree under different intensities of rainforest canopy coverage. Cerne 2019, 25, 93−104.
  • Moretti, A.P.; Olguin, F. Y.; Pinazo, M.A.; Gortari, F.; Bahima, J.V.; Graciano, C. Supervivencia y crecimiento de un árbol nativo maderable bajo diferentes coberturas de dosel en el Bosque Atlántico, Misiones, Argentina. Austral 2019, 29, 99−111.
  • Olguin, F.Y.; Moretti, A.P.; Pinazo, M.; Gortari, F. Bahima, J.V.; Graciano, C. Morphological and physiological plasticity in seedlings of Araucaria angustifolia and Cabralea canjerana is related to plant establishment performance in the rainforest. Forest Ecol. Manag. 2020, 460, 117867.
  • Morales, A.; Kaiser, E. Photosynthetic acclimation to fluctuating irradiance in plants. Front. Plant Sci. 2020, 11, 268. https://doi.org/10.3389/fpls.2020.00268
  • Poorter, H.; Niinemets, Ü.; Ntagkas, N.; Siebenkäs, A.; Mäenpääs, M.; Matsubara, S.; Pons, T.L. A meta-analysis of plant responses to light intensity for 70 traits ranging from molecules to whole plant performance

Reviewer comment: They mentioned that it is interesting to study additional species:

That is merely their opinion, and the reader (including me) do not think the present manuscript is interesting. (and I rarely found an expression like “it is interesting” in high-impact journals). The reason of this is that they do not explain what is the novel and important points of their study. Every dataset is novel, but they should explain why it is important and novel. This point is particularly important to be published in high-impact journals.

Response: Thanks for sharing your point of view. Our opinion of qualifying as “interesting” the study of acclimation responses in a larger number of species was based on the evidence supporting our experimental approach, because according to the literature these aspects deserve attention, especially for the implications that this kind of knowledge could have in the sustainable management of natural resources. We have modified the term to avoid give a personal connotation as argued by the reviewer; however, we agree to certain extent because we feel that this point was judged from a perspective in the same sense.

Reviewer comment: Additionally, I wonder whether there are really no study on the sun vs shade acclimation of the two species (C. alata and E. cyclocarpum).

Response: Our research question was formulated because we identified a gap of knowledge into the topic. The reference given below, although rather important, aims other research question.

Foroughbakhch et al. Forest Ecology and Management Volume 235, Issues 1–3, 1 November 2006, Pages 194-201, https://doi.org/10.1016/j.foreco.2006.08.012

Reviewer comment: Methodology is poorly described.They set PPFD levels (25, 35, 55, 70) using films. However, if my understandings is correct, these values are the transmissivity of the plastic film, which may not necessary matches the actual PPFD experienced by the study plants. They mentioned that they have actually measured PPFD. If so, they should show the actual rPPFD values experienced by the plants.

Response: We appreciate the comment, but we also believe that the opinion issued does not correspond to the corrections that the work deserves. Both, as it is carried out in several studies and as it is explained within the text, we define the light treatments based on several measurements of the photosynthetic photon flux density (PPFD) above and below of the shade cloths to determine the exact level of transmissivity or shading. We present the light levels in terms of percentage of PPFD as a more precise measurement of the light environment prevailing in each light condition tested, independently of the fluctuation in irradiance that instantaneous measurements of PPFD usually record.  This methodological approach guarantees that the plants receive only the amount of light that the nets allow according to their values of transmissivity. This is useful because offers a practical basis that can be reproducible in future studies. Moreover, we preferred this approach because we pretend that our results can be applied for improving or optimizing the nursery or field management from an operational perspective. Poorter et al. (New Phytologist, 2019, 223: 1073–1105) mention that instantaneous values of PPFD are less informative, as light intensity varies strongly both diurnally and among days, therefore, instead, they strongly recommend that DLI is measured for the duration of an experiment. This is the reason why we also report values of light in DLI, which is the PPFD integrated over a day (mol m-2 d-1 ).

Reviewer comment: (Table 5) Information on daily integral of outside (open-sky condition) should also be provided.

Response: The information is provided within the text.

Reviewer comment: In several recent studies it has been recognized that using shade cloth or film may not always mimic the natural light environment in shaded understory, because shade film or cloth cannot produce sunflecks (see literatures listed below). This fact has long been recognized (e.g., Valladares et al. 2008; Poorter et al. 2016; Morales & Kaiser 2020), and recent studies show quantitative evidence (Deguchi & Koyama 2020, listed above). I think the data presented here is potentially valuable, but I recommend the authors to clarify the limitation of the present results obtained from artificial shading experiment, with an argument on this point.

Response: This opinion raises an interesting point of view. Certainly, the simulation of the light environment that occurs in natural conditions remains a challenging due to fluctuation in irradiance because of different sources of fluctuation, one of them sunflecks. Sunflecks, undoubtedly, according to literature have a strong influence on plant acclimation responses.   However, as we argue along the study, we have formulated our research question in another aspect inherent to light, changes in its availability in terms of amount. In that sense, the methodological approach was designed to get responses about our research question and the conclusions that we present are consistent with the findings and arguments that are addressed into the topic. We have no experimental evidence that sustains possible limitations for not simulating natural conditions; however, we state that in future studies about acclimation responses of plant to heterogeneous light environments, the influence of sunflecks should be examined to get a more comprehensive understanding on plant functioning to environmental constraints.

Round 2

Reviewer 2 Report

The authors have improved the manuscript according to most of my suggestions. I still have a minor comment shown below. I would recommend acceptance of this manuscript on the condition that the following remaining point is addressed.

I found that some literatures relevant to the present study, which I have suggested in my previous comment, are not discussed in the revised manuscript (I might have missed the Deguchi & Koyama in the literature list in the previous comment (shown below)), which is I think is relevant to the Introduction and limitation (i.e., using shade cloth vs natural light) in the present manuscript).

In several recent studies it has been recognized that using shade cloth or film may not always mimic the natural light environment in shaded understory, because shade film or cloth cannot produce sunflecks. This fact has long been recognized (e.g., Valladares et al. 2008; Poorter et al. 2016; Morales & Kaiser 2020), and recent studies show quantitative evidence (Deguchi & Koyama 2020). I think the data presented here is potentially valuable, but I recommend the authors to clarify the limitation of the present results obtained from artificial shading experiment, with an argument on this point.

Additionally, although I appreciate that the aim of the study by Foroughbakhch et al. are not exactly the same from the present study, this would not be ignored because Foroughbakhch et al. also investigated growth of the same two study species, aiming at applications in reforestation.

Suggested Literatures

  1. Deguchi & Koyama, Forests 2020, 11(12), 1365; https://doi.org/10.3390/f11121365
  2. Foroughbakhch et al. Forest Ecology and Management Volume 235, Issues 1–3, 1 November 2006, Pages 194-201, https://doi.org/10.1016/j.foreco.2006.08.012

Author Response

Reviewer comment: In several recent studies it has been recognized that using shade cloth or film may not always mimic the natural light environment in shaded understory, because shade film or cloth cannot produce sunflecks (see literatures listed below). This fact has long been recognized (e.g., Valladares et al. 2008; Poorter et al. 2016; Morales & Kaiser 2020), and recent studies show quantitative evidence (Deguchi & Koyama 2020, listed above). I think the data presented here is potentially valuable, but I recommend the authors to clarify the limitation of the present results obtained from artificial shading experiment, with an argument on this point.

Response: We really appreciate reviewer remarks on this point and his/her willingness to feedback our manuscript. We have reconsidered our point of view to address the modifications requested.   In this way, we argue that given that artificial conditions, as those generated by using shade cloths, cannot simulate a fluctuating environment of irradiance as naturally occurs in the understory (Deguchi & Koyama 2020), we, as recommended by the reviewer, state possible limitations that our results could have to be aware readers about the scope of our findings. Therefore, we explain within the text that further studies on this topic need to consider the influence of light fluctuations, as those generated due to sunflecks, in order to understand in greater detail all the possible responses of acclimation and their interactions or trade-offs, that plants undergo in a fluctuating irradiance environment because this represents a constrains to which plants should modulate themselves a set of morphological and physiological features, depending on the prevailing environment, so they be able to attain maximum photosynthetic rates and an appropriate behavior of other plant functions. Possibly, in face of the methodological limitation that implies not evaluating under similar conditions than those occurring in nature, the studied plants did not exhibit all their potential of acclimation in the variety of light conditions tested.

Reviewer comment: Additionally, although I appreciate that the aim of the study by Foroughbakhch et al. are not exactly the same from the present study, this would not be ignored because Foroughbakhch et al. also investigated growth of the same two study species, aiming at applications in reforestation.

Response: We have considered this reference as a source of information to support species suitability to be used in reforestations or restoration projects.

Round 3

Reviewer 2 Report

The authors have succesfully modified the manuscript according to all of my suggestions. Therefore, I now reccomend publication of this manuscript.